# 17-AAG-Induced Activation of the Autophagic Pathway in *Leishmania* Is Associated with Parasite Death

**DOI:** 10.3390/microorganisms9051089

**Published:** 2021-05-19

**Authors:** Antonio Luis de O. A. Petersen, Benjamin Cull, Beatriz R. S. Dias, Luana C. Palma, Yasmin da S. Luz, Juliana P. B. de Menezes, Jeremy C. Mottram, Patrícia S. T. Veras

**Affiliations:** 1Laboratory of Parasite-Host Interaction and Epidemiology (LAIPHE), Gonçalo Moniz Institute—FIOCRUZ, Salvador 40296-710, Brazil; petersen.swe@gmail.com (A.L.d.O.A.P.); beatrizrsdias@gmail.com (B.R.S.D.); luanabio@hotmail.com (L.C.P.); yasmindasilvaluz.1997@gmail.com (Y.d.S.L.); juliana.fullam@fiocruz.br (J.P.B.d.M.); 2Wellcome Trust Centre for Molecular Parasitology, Institute of Infection, Immunity and Inflammation, College of Medical Veterinary and Life Sciences, University of Glasgow, Glasgow G12 8QQ, UK; bcull53@hotmail.com (B.C.); jeremy.mottram@york.ac.uk (J.C.M.); 3Post-Graduate Program in Experimental Pathology, Federal University of Bahia (UFBA), Salvador 40110-909, Brazil; 4Department of Biology, York Biomedical Research Institute, University of York, York YO10 5DD, UK; 5National Institute of Tropical Disease (INCT-DT), National Council for Scientific and Technological Development (CNPq), Brasília 71605-001, DF, Brazil

**Keywords:** Hsp90, leishmaniasis, chemotherapy, Hsp90 inhibitors, autophagy, ubiquitin

## Abstract

The heat shock protein 90 (Hsp90) is thought to be an excellent drug target against parasitic diseases. The leishmanicidal effect of an Hsp90 inhibitor, 17-N-allylamino-17-demethoxygeldanamycin (17-AAG), was previously demonstrated in both in vitro and in vivo models of cutaneous leishmaniasis. Parasite death was shown to occur in association with severe ultrastructural alterations in *Leishmania*, suggestive of autophagic activation. We hypothesized that 17-AAG treatment results in the abnormal activation of the autophagic pathway, leading to parasite death. To elucidate this process, experiments were performed using transgenic parasites with GFP-ATG8-labelled autophagosomes. Mutant parasites treated with 17-AAG exhibited autophagosomes that did not entrap cargo, such as glycosomes, or fuse with lysosomes. ATG5-knockout (Δ*atg5*) parasites, which are incapable of forming autophagosomes, demonstrated lower sensitivity to 17-AAG-induced cell death when compared to wild-type (WT) *Leishmania*, further supporting the role of autophagy in 17-AAG-induced cell death. In addition, Hsp90 inhibition resulted in greater accumulation of ubiquitylated proteins in both WT- and Δ*atg5*-treated parasites compared to controls, in the absence of proteasome overload. In conjunction with previously described ultrastructural alterations, herein we present evidence that treatment with 17-AAG causes abnormal activation of the autophagic pathway, resulting in the formation of immature autophagosomes and, consequently, incidental parasite death.

## 1. Introduction

*Leishmania* spp. are protozoan parasites [1,2] that cause leishmaniasis, which can present in a variety of clinical manifestations, including skin and visceral forms [3]. Being one of the most important neglected tropical diseases, leishmaniasis affects millions of people worldwide. Financial investment in new therapeutic strategies has been scarce [3], resulting in pentavalent antimonials being the drug of choice for more than 70 years in Brazil. However, antimonial therapy requires parenteral administration at high dosage and involves a lengthy therapeutic course that can result in a range of serious side effects [4]. Recent increases in therapeutic failure [5,6,7] reinforce the importance of developing new drugs capable of replacing or complementing existing strategies for leishmaniasis treatment.

Heat shock protein 90 (Hsp90) has been considered as a potential molecular target for the treatment of parasitic diseases [8,9,10]. Hsp90 inhibitors, such as geldanamycin or 17-N-allylamino-17-demethoxygeldanamycin (17-AAG), have demonstrated inhibitory effects on the differentiation process of *Leishmania* in vitro [11] and were shown to exert anti-parasitic activity in vitro and in vivo [12,13,14,15,16]. These inhibitors are members of a family of antibiotics that selectively bind to the Hsp90 ATP pocket, preventing ATP hydrolysis and folding of client proteins that do not achieve a tertiary structure. In mammals, these unfolded proteins are eventually degraded in the ubiquitin-proteasome system, which can result in cell death secondary to proteasome overload. This can subsequently lead to the formation of protein aggregates [17,18,19,20], resulting in the activation of a protective selective autophagic process in order to avoid aggregate accumulation in the cytoplasm [21,22,23]. Alternatively, Hsp90 inhibition can lead to a pronounced transcription of Hsp70, Hsp90 and Hsp40, responsible for mounting mis- or unfolded proteins, thereby limiting the formation of polyubiquitylated protein aggregates [24].

In previous studies, we have demonstrated that 17-AAG was capable of controlling *Leishmania* infection (in vitro [15] and in vivo [16]) by eliminating promastigotes, which colonize the insect vector, as well as amastigotes, which are found within vertebrate host cells [15,16]. Nevertheless, the mechanism by which Hsp90 inhibition causes parasite death remains unclear. Electron microscopy revealed ultrastructural alterations suggestive of the activation of autophagy in parasites, including progressive cytoplasmic vacuolization, double-membrane vacuoles, myelin figures and vacuoles containing cytoplasmic material, all occurring in the absence of significant alterations in cellular nuclei, mitochondria or plasma membranes [15].

The conserved autophagic process in eukaryotic cells is responsible for the turnover of long-lived proteins and organelles inside autophagosomes [25,26], which plays an important role in cellular homeostasis and in cell survival in response to different types of stress [25,27,28,29]. Autophagosomes are formed in successive steps involving the recruitment and activation of proteins of the ATG (AuTophaGy-related genes) family [30,31,32]. In *Leishmania* parasites, ATG12 must firstly conjugate with ATG5 in order for ATG8 to participate in the assembly of this complex, resulting in the formation of autophagosomes [33,34,35] that may acquire cargo and fuse with lysosomes, thereby forming autolysosomes [33,34]. The engulfed material is degraded, generating small molecules that may be utilized for cell survival [36,37]. Autophagy has also been identified as essential to the differentiation of *Leishmania* promastigotes into amastigotes [33]. By contrast, autophagic induction has been associated with death in eukaryotic cells [30,38]. Thus, the true role played by autophagy with respect to the mechanism responsible for causing protozoan parasite death in response to several stress stimuli, including antiparasitic drugs, remains to be elucidated [39].

We hypothesize that 17-AAG induces abnormal activation of autophagy in *Leishmania* spp., resulting in parasite death. To test this, several genes of the autophagic pathway were genetically modified in *L. major* promastigotes, which were used to investigate the participation of autophagy in parasite death following treatment with 17-AAG.

## 2. Materials and Methods

### 2.1. Leishmania Culturing

*Leishmania major* (MHOM/JL/80/Friedlin) were cultivated in modified HOMEM medium (Gibco, Carlsbad, CA, USA) supplemented with 10% (*v/v*) heat-inactivated fetal calf serum (Gibco) and 1% (*v/v*) penicillin streptomycin solution (Sigma, St. Louis, MO, USA) at 25 °C until mid-log phase was achieved, corresponding to 5 × 10^6^ parasites/mL.

### 2.2. Generation of Parasite Mutants Expressing Fluorescent Markers

All mutant parasites used in this research were previously generated by our collaborators and are described as follows: (i) green fluorescent protein-ATG8 (GFP-ATG8) plasmid as described by Besteiro et al. [33]; (ii) the glycosome targeting SQL motif labelled with RFP plasmid (RFP-SQL) as described by Cull et al. [40]; (iii) the proCPB lysosomal-marker labeled with RFP plasmid (RFP-proCPB) by Huete-Perez et al. [41]. Null mutant *atg5* parasites (Δ*atg5*), Δ*atg5* expressing GFP-ATG8 (Δ*atg5*(GFP-ATG8)) and Δ*atg5*::ATG5 parasites re-expressing ATG5 in the Δ*atg5* null mutant were generated by Williams et al. [35] and used as controls. In sum, two plasmids, both derived from pGL345-HYG, the pGL345ATG5-HYG5′ 3′ and pGL345ATG5-BLE5′ 3′, were generated with fragments of the 5′ and 3′ UTRs flanking the ORF of ATG5 gene. The resulting linearized cassettes were used in two rounds of electroporation using a nucleofector transfection system according to the manufacturer’s instructions (Lonza, Basel, Switzerland) to produce a heterozygous cell line, simultaneously resistant to hygromycin and bleomycin. To select the parasites that successfully expressed the desired proteins, an appropriate antibiotic was used to treat each transfected parasite line: G418 (Neomycin) at 50 μg/mL; Hygromycin at 50 μg/mL; Blasticidin S at 10 μg/mL; Phleomycin at 10 μg/mL (all from InvivoGen, San Diego, CA, USA).

### 2.3. Parasite Treatment with 17-AAG or Pentamidine

In accordance with each experimental protocol, promastigotes of *L. major* were submitted to treatment procedures using the antileishmanial 17-AAG (InvivoGen, San Diego, CA, USA) (100, 300 or 500 nM) or pentamidine (Sigma, St. Louis, MO, USA) (10, 20 and 30 μM) for up to 72 h. At the end of each treatment period, parasites were pelleted by centrifugation for 3 min at 1000× *g* and then washed thrice in PBS for medium removal. Cells were then resuspended in PBS and a 10 μL suspension was spread thinly over a slide covered with a 22 × 40 mm coverslip, then sealed with nail varnish to perform fluorescence microscopy.

### 2.4. Assessment of Autophagosome Formation and Autophagosome Colocalization with Glycosomes and Lysosomes by Fluorescence Microscopy

Fluorescence microscopy was used: (i) to assess the presence of GFP-ATG8-labelled vesicles, characteristic of autophagosomes, which appear as punctate structures as previously described [40,42] and (ii) to evaluate the effects of 17-AAG treatment on the autophagosomal maturation process using the two *L. major* double-mutant promastigotes: GFP-ATG8 and RFP-SQL; GFP-ATG8 and proCPB-RFP. To quantify the number of GFP-ATG8-labelled vesicles and determine the percentage of parasites containing GFP-ATG8-labelled vesicles, after mounting for up to 1 h, parasite smears were observed under a DeltaVision Core deconvolution microscope (Applied Precision Inc., Issaquah, WA, USA) using GFP or mCherry filters at 1,000 magnification. For cell visualization, DIC images were also obtained under polarized light. All images were processed and analyzed using SoftWoRx image analysis software (Applied Precision Inc.) [40,42]. At least 300 parasites were counted for each combination of treatment and time point, with no less than three independent experiments considered. Also, the effects of 17-AAG treatment on the autophagosomal maturation process was evaluated in the double-mutant *L. major* promastigotes. Autophagosome colocalization with glycosomal cargo or with lysosomes was evaluated after treatment with 500 nM of 17-AAG at 24 or 48 h. Parasites were also imaged using a DeltaVision Core deconvolution microscope as described above. Images were submitted to colocalization analysis using SoftWoRx image analysis as previously described [43].

### 2.5. Parasite Viability

Axenic parasites in mid-log phase were treated with serial dilutions of 17-AAG for 48 h at concentrations ranging from 10 to 15,625 nM or with DMSO (control), and after adding AlamarBlue (Invitrogen, Carlsbad, CA, USA) (10% final concentration) followed by incubation for another 24 h at 24 °C, reagent absorbance was measured at the wavelengths of 570 and 600 nm.

To evaluate the kinetics of 17-AAG on the viability of Δ*atg5* compared to WT and Δ*atg5*::ATG5 parasites, promastigote cultures were treated with 300 nM and 500 nM of the Hsp90 inhibitor for 24 h, 48 h and 72 h at 24 °C. AlamarBlue was added and absorbance was measured as described above. Treatment effect was estimated by determining the area under the curve (AUC) for each group.

### 2.6. Parasite Growth Curve

*Leishmania* cell lines (WT, Δ*atg5* and Δ*atg5*::ATG5) at a concentration of 10^5^ cells/mL were incubated in 10 mL of HOMEM medium supplemented with 10% (*v/v*) heat-inactivated fetal calf serum and 1% (*v/v*) penicillin streptomycin solution. Cells were treated or not with 100 nM of 17-AAG and counts were performed daily for at least 13 days. Parasite numbers were recorded and plotted.

### 2.7. Western Blot to Assess Ubiquitylated Proteins

The accumulation of ubiquitylated proteins in parasites treated with 17-AAG (500 nM) for 24 h was evaluated by Western blot analysis. As a positive control, parasites were treated with the proteasome inhibitor MG132 (3 μM) for 24 h. Following treatment, parasites were pelleted by centrifugation for 3 min at 1000× *g*, and then washed thrice in PBS for medium removal. Protein extraction was then performed for Western blot analysis as described below.

Parasites were lysed with laemmli buffer (2-Mercaptoethanol 0.1%, bromophenol blue 0.0005%, glycerol 10%, SDS 2%, Tris-HCl 63 mM, pH 6.8) at a proportion of 10 μL of buffer for 10^6^ parasites, then boiled for 5 min and cell extracts were stored at −20 °C. Proteins were transferred from a 12% polyacrylamide gel, following electrophoresis, to a Hybond-C nitrocellulose membrane (Amersham, GE Healthcare, Little Chalfont, UK). Transfer was carried out by semi-dry blotting using a BioRad Trans-Blot SD Semi-Dry Transfer Cell at 30 volts for 45 min, with membranes and filter papers soaked in transfer buffer (20 mM Tris-HCl, 15 mM glycine, 20% (*v/v*) methanol, in distilled water). Membranes were subsequently incubated in a blocking solution for 1 h at room temperature or overnight at 4 °C under agitation. After blocking, each membrane was incubated with 1:1000 FK2 anti-ubiquitin antibody (LifeSensors, Malvern, PA, USA) diluted in fresh TBST buffer with 3% milk for 1 h at room temperature. Secondary anti-mouse antibody conjugated with horseradish peroxidase (HRP) (Promega, Madison, WI, USA) was diluted at 1:10,000 in fresh TBST buffer and each membrane was first incubated with an ECL (Enhanced Chemiluminescence) solution (SuperSignal West Pico Chemiluminescent Substrate Kit, Pierce, Rockford, IL, USA), and then exposed on Kodak photographic film. An antibody against elongation factor 1α (EF1α) (Millipore, Germany) was used for loading control. Western blotting experiments were performed three times.

### 2.8. Assessment of Protein Aggregation

Axenic promastigotes of *L. major* lines (WT, Δ*atg5* and Δ*atg5*::ATG5) were treated with 17-AAG (500 nM) or MG132 (3 μM) for 24 h at 24 °C. The aggregation of soluble proteins was analyzed following a previously described protocol [44] employing SDS-PAGE. Briefly, 3 × 10^8^ parasites were centrifuged at 1800× *g* at 4 °C for 10 min. The pellet was resuspended in 500 μL of lysis buffer (50 mM potassium phosphate buffer, 1 mM EDTA, 5% glycerol, 1 mM protease inhibitor, Roche, Mannheim, Germany) and submitted to a 5 freeze-thaw cycles (liquid nitrogen-water). Intact cells were removed by centrifugation at 1800× *g* for 5 min and proteins were quantified. Then, 1 mg of protein from each group was centrifuged at 15,000× *g* for 20 min to isolate the membrane and aggregate fractions. Next, pellets were resuspended in lysis buffer, then sonicated and membrane proteins were removed via the addition of 2% NP40 followed by centrifugation at 15,000× *g* for 20 min. Pellets were resuspended in 100 μL of SDS sample buffer and heated at 95 °C for 5 min. Samples were then analyzed using 10% SDS-PAGE, followed by silver labeling using a Bio-Rad Silver Stain kit (Bio-Rad, Hercules, CA, USA) in accordance with the manufacturer’s instructions. Experiments were independently repeated three times.

### 2.9. Statistical Analysis

The half maximal inhibitory concentration (IC_50_) of 17-AAG in *L. major* WT, Δ*atg5* and Δ*atg5*::ATG5 promastigotes was determined by performing sigmoidal regression on each respective concentration-response curve. Data are presented as the mean ± standard deviation of the mean under parametric analysis (One-way ANOVA or Welch’s ANOVA test followed by Tukey’s or Dunnett’s Multiple Comparisons) or medians and quartile ranges in the case of non-parametric analysis (Kruskal-Wallis test, Dunn’s multiple comparison). All data were analyzed using the Prism program (GraphPad software, V. 9.1.0 La Jolla, CA, USA).

## 3. Results

### 3.1. 17-AAG Induces Autophagosome Formation in Promastigote Forms of Leishmania

Treatment of *Leishmania* parasites with 17-AAG at the concentration of 500 nM resulted in an increased number of parasites containing green-labeled punctate structures (Figure 1A). After 48 h of treatment at both 300 and 500 nM, among counted parasites, 33.1% (Q1: 29.3; Q3: 33.9) and 37.4% (Q1: 34.7; Q3: 37.8, *p* < 0.05), respectively, of parasites contained punctate structures, while these alterations were seen in only 19.2% (Q1: 17.2; Q3: 21.1) of control parasites (Figure 1B; *p* < 0.05). This percentage of *L. major* promastigotes containing punctate structures increased after 72 h of treatment, with a median value of 36.1% (Q1: 32.1; Q3: 44.4) of the parasites treated with 300 nM and 50.2% (Q1: 43.5; Q3: 55.2, *p* < 0.05) of those treated with 500 nM revealing labeled vesicles in the cytosol, while the level of control parasites containing punctate structures was 20.1% (Q1: 18.3; Q3: 22.6) at 72 h of treatment (Figure 1B). The median values of the number of punctate structures in parasites treated with 300 nM of 17-AAG at 48 and 72 h were not statistically different compared to diluent-treated *L. major* promastigotes. The median number of GFP-ATG8-labeled vesicles per parasite after 48 and 72 h was higher in parasites treated with 500 nM of 17-AAG, with values of 1.8 (Q1: 1.7;Q3:1.8, *p* < 0.05) and 2.0 (Q1: 1.8; Q3: 2.2), respectively, compared to median values of 1.4 (Q1: 1.2; Q3: 1.5, *p* < 0.05) and 1.2 (Q1: 1.2; Q3: 1.4) (Figure 1B) in control parasites. The atg5-deficient parasites expressing GFP-ATG8 (Δ*atg5*(GFP-ATG8)) treated with 17-AAG exhibited no fluorescent punctate structures, confirming that the vesicles detected in GFP-ATG8 parasites were indeed autophagosomes (Figure 1C). As an additional control, parasites were treated with another antileishmanial, pentamidine, at concentrations of 10, 20 or 30 μM for 24, 48 and 72 h. Treatment with pentamidine caused cell death (data not shown), yet few punctate structures were detected in the cytosol of treated parasites (Figure 1D). This indicates that parasite death is not always associated with autophagosome formation in *Leishmania* parasites.

### 3.2. 17-AAG Inhibits the Autophagosome Maturation Process

Since the inhibition of endosome and autophagosome fusion with lysosomes by chloroquine has been previously shown to be involved in parasite death [45,46,47], we investigated whether treatment with 17-AAG was capable of altering the autophagosomal maturation process. In double-mutant parasites expressing GFP-ATG8 and the glycosomal marker, RFP-SQL, a mean value of 22.7% ± 4.4 of the total number of counted autophagosomes colocalized with glycosomes in parasites treated with 17-AAG (500 nM, 24 h), similar to untreated double-mutant parasites (30.5% ± 6.4) (Figure 2A,B). After 48 h of treatment, a comparable proportion of colocalization was observed in labeled compartments (20.5% ± 3.9) in treated parasites, which was significantly lower than that detected in control parasites treated with DMSO, 41.6% ± 5.2 (Figure 2A,B; *p* = 0.0006). Additionally, double-mutant parasites expressing both GFP-ATG8 and the lysosomal marker, proCPB-RFP, when treated with 17-AAG (500 nM) for 24 and 48 h, exhibited a remarkably lower proportion of autophagosome-lysosome colocalization after 24 h, with mean values of 8.2% ± 5.1 (*p* = 0.0197) and 48 h, 12.1% ± 11.6, in comparison to DMSO-treated controls: 35.6% ± 16.6 and 38.6% ± 25.6, respectively (Figure 2C,D). These findings indicate that 17-AAG treatment resulted in inhibition of the autophagosome maturation process via fusion inhibition of ATG8-labelled vesicles with compartments labelled with lysosomal and glycosomal markers.

### 3.3. atg5-Deficient Parasites Are More Resistant to 17-AAG-Induced Cell Death Than WT Parasites

To evaluate whether autophagy plays a role in 17-AAG-induced parasite death, IC_50_ values were determined for Δ*atg5* parasites treated with 17-AAG for 48 h, which showed a mean value of 174.3 nM ± 15.7, 83.7% higher than that determined for WT (Figure 3A, *p* < 0.01). Moreover, when the *atg5* gene was added back to the Δ*atg5* parasites (Δ*atg5*::ATG5), the resulting IC_50_ value was 104.3 nM ± 32.9, similar to that found in WT *L. major* promastigotes (95.0 nM ± 23.1) (Figure 3A, *p* < 0.01). In addition, when cultivated in medium containing 100 nM of 17-AAG for 13 days, Δ*atg5* parasites grew faster than either WT or Δ*atg5*::ATG5, as assessed by growth curves (Figure 3B) and the area under the curve (AUC) (Figure 3C). This marked growth was especially noticeable during the log growth phase (days 5–6) when 500% more Δ*atg5* parasites were seen compared to WT or Δ*atg5:*:ATG5 (Figure 3B). In contrast, no differences in parasite growth rates were observed among these three parasite lines when cultivated in 17-AAG-free medium (Figure 3B). Even when the highest concentrations of 17-AAG (300 and 500 nM) were administered to promastigotes for up to 72 h, less toxicity was evidenced in the Δ*atg5* lineage compared to WT or Δ*atg5:*:ATG5 *L. major* parasites (Figure 3D). These findings provide evidence that Δ*atg5* parasites are less susceptible to cell death following treatment with 17-AAG, which suggests the participation of autophagy in inhibitor-induced parasite death.

### 3.4. 17-AAG Treatment Results in Increased Accumulation of Ubiquitylated Proteins, But Not Protein Aggregates, in L. major Parasites

Due to the participation of autophagy in *Leishmania* death arising from 17-AAG treatment, we hypothesized that autophagic activation could be consequent to Hsp90 inhibition, which causes a subsequent enhancement in the amount of ubiquitylated protein. Low basal levels of ubiquitylated proteins were seen in all untreated WT, Δ*atg5* and Δ*atg5*::ATG5 parasites (Figure 4). Treatment with 17-AAG (500 nM for 48 h) induced an overall increase in the amounts of ubiquitin-labeled proteins in WT, Δ*atg5* and Δ*atg5*::ATG5 parasites (Figure 4). A similar result was observed after 48 h of 17-AAG treatment (data not shown). As expected, of the three lines evaluated, Δ*atg5* parasites demonstrated the greatest accumulation of ubiquitylated proteins after treatment with either 500 nM of 17-AAG or 3 μM of MG132 (Figure 4). Predictably, the treatment of parasites with MG132 (3 μM, 48 h) increased the accumulation of ubiquitylated proteins in all three parasite lines evaluated. Moreover, treatment with MG132 also resulted in a higher proportion of autophagosomes in GFP-ATG8 parasites, as evidenced by the mean percentage of parasites bearing punctate structures: 14.7% ± 3.5 in parasites treated for 24 h with 3 μM of MG132, in comparison to a mean value of 10.9% ± 1.5 in control parasites (Figure 5A,B). This difference increased at 48 h to 30.6% ± 4.6 of parasites treated with 3 μM of MG132 compared to 6.8% ± 2.0 in controls (Figure 5B, *p* = 0.0007). As was also expected, positive controls treated with 500 nM of 17-AAG exhibited a significant increase in the percentage of parasites containing labeled vesicles: 20.8% ± 1.3 (*p* = 0.0035) at 24 h and 21.1% ± 4.3 at 48 h (*p* = 0.0091) (Figure 5B). Increased ubiquitylated protein accumulation can result in proteasomal overload and the accumulation of protein aggregates or, alternatively, enhancement in the transcription of Hsp70, Hsp90 and Hsp40 [48].

Using SDS-PAGE, protein extracts of all parasite strains: WT, Δ*atg5* and Δ*atg5*::ATG5 treated with 17-AAG (500 nM for 24 h) did not result in increased protein aggregate formation in comparison to untreated parasites, while positive control parasites treated with MG132 (3 µM for 24 h) revealed increased amounts of protein aggregates (Figure 6).

## 4. Discussion

The present study confirmed that in *Leishmania* treated with 17-AAG, autophagy is induced by an increased percentage of autophagosomes expressing GFP-ATG8, as well as higher overall numbers of labeled autophagosomes per parasite. We also found that the macroautophagy-deficient Δ*atg5*[GFP-ATG8] *Leishmania*, which is unable to form autophagosomes, did not form any detectable punctate structures.

The present study also found that treatment with 17-AAG induced a reduction in the degree of colocalization between autophagosomes and glycosomes, as well as between autophagosomes and lysosomes, in comparison to controls. Hsp90 is known to control the expression of hundreds of proteins involved in diverse cell functions [49,50]. A previous report described the involvement of Hsp90 in controlling vesicle trafficking and fusion by folding proteins, responsible for recycling RAB proteins from vesicle membranes back into the cytoplasm [51] and vesicle transport proteins that play a role in the transport of glycoproteins from the Golgi to other compartments [52]. Since reduced colocalization of the proCPB-RFP and GFP-ATG8 was observed in *Leishmania* following treatment with 17-AAG compared to untreated parasites, we speculate that Hsp90 inhibition results in the unfolding or incorrect folding of parasite proteins involved in vesicle trafficking and fusion. The inhibition of the fusion of newly-formed autophagosomes with lysosomes could result in the trapping of proteins and organelles within autophagosomes, leading to parasite death. In agreement with our findings, a recent study described that disrupting a cysteine protease located in the vacuolar compartment (VAC) of *Toxoplasma gondii* caused a reduction in the proteolytic activity of parasite lysosomes, the accumulation of undigested autophagosomes in parasite cytoplasm, and subsequently, a reduction in the intensity of infection [53].

We found that survival in Δ*atg5 L. major* promastigotes increased compared to WT parasites under treatment with 17-AAG. Indeed, we showed that Δ*atg5 L. major* were not only able to survive and proliferate more efficiently than WT at a low dosage of 17-AAG (100 nM) for 13 days, but were also found to be more resistant to death at higher dosages (300 and 500 nM). These findings lead us to propose that the activation of the autophagic pathway contributes to *Leishmania* cell death. Macroautophagy is a successful adaptive strategy that functions as a protective mechanism activated under different physiological stress stimuli [54,55,56]. Similarly to our study, it has been shown that autophagy is induced in *T. gondii* in response to endoplasmic reticulum stress, followed by the accumulation of unfolded proteins [57]. Also, incomplete autophagosome maturation was shown to be harmful to eukaryotic cells [58], including mammals [48,59] and *T. gondii* [53]. Moreover, it was previously demonstrated that several *Leishmania* lines present an inability to complete the transformation process from promastigotes to amastigotes, including *Δatg5* parasites [35], *Δatg4.2* parasites expressing a mutant vesicular sorting protein 4 [33] form autophagosomes that do not fuse with lysosomes, and *Δcpa*/*cpb* parasites, which do form autophagosomes that fuse with lysosomes, produce non-degraded lysosomal content due to the deficiency of CPA and CPB enzymes [42,60]. This inability to complete transformation leads to a reduced survival rates inside macrophages in vitro [33,35,42,60] and in vivo [35].

The inhibition of Hsp90 in cancer cells results in an increase in the accumulation of ubiquitylated proteins in the cytosol [19], and subsequently, proteasome overload, leading to both the accumulation of unfolded and misfolded proteins [23,61] and protein aggregate formation [62,63,64]. We speculate that a similar event could take place in parasites treated with 17-AAG. The accumulation of polyubiquitylated proteins following treatment with Hsp90 inhibitors in animal models of neurodegenerative disease [17,24,65] inhibited protein aggregate formation from the activation of Hsp70 and Hsp40 [48,66]. The present study showed that treating *Leishmania* parasites with 17-AAG led to the accumulation of ubiquitylated proteins at levels similar to those observed in parasites treated with the proteasome inhibitor, MG132. Although 17-AAG treatment induced the accumulation of polyubiquitylated proteins, Hsp90 inhibition most likely did not result in proteasome overload, as no enhancement in the formation of protein aggregates was detected (Figure 6), likely due to the activation of other Hsps [66].

In sum, our findings evidence that *Leishmania* cell death caused by 17-AAG is associated with abnormal activation of the autophagic pathway, resulting in the formation of autophagosomes unable to achieve complete autophagolysosomal maturation and therefore incapable of degrading engulfed material.

## Figures and Tables

**Figure 1 microorganisms-09-01089-f001:**
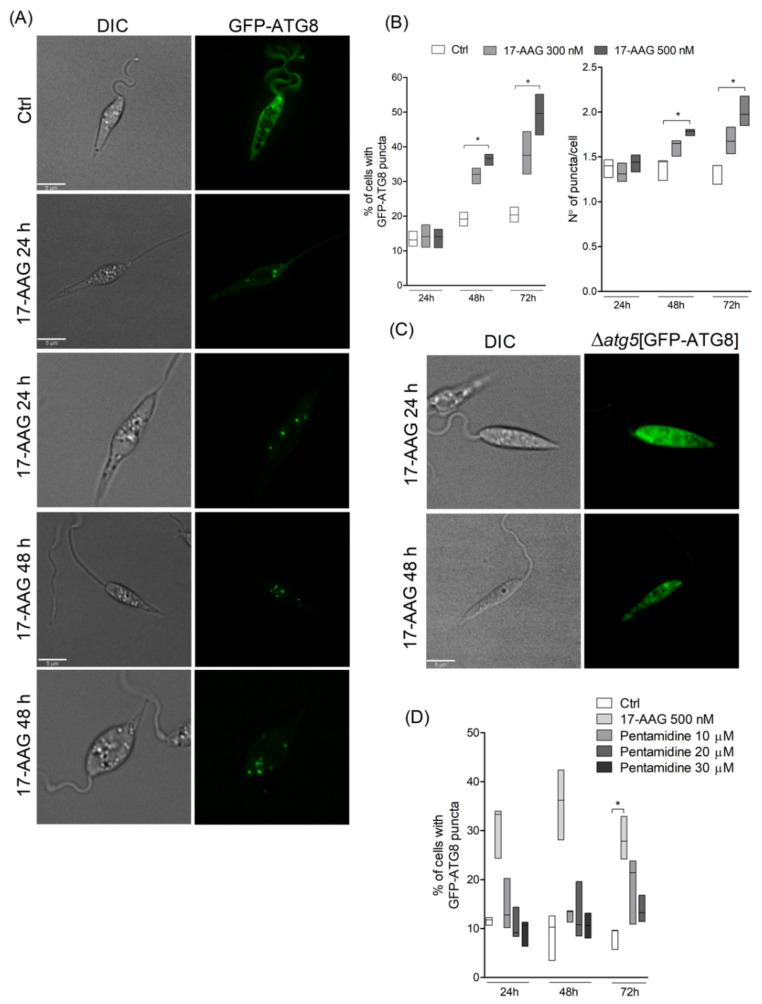
Evaluation of autophagosome formation in *Leishmania* promastigotes following treatment with 17-AAG. (**A**) Axenic promastigotes of *Leishmania* expressing GFP-ATG8 were treated or not with 17-AAG (500 nM) for 24 or 48 h and imaged by fluorescence microscopy. (**B**) The percentage of cells bearing autophagosomes and the number of autophagosomes per cell were calculated at 24, 48 and 72 h after treatment with 17-AAG (300 or 500 nM). (**C**) Δ*atg5*[GFP-ATG8] parasites were treated with 17-AAG and imaged by fluorescence microscopy. (**D**) Comparison of the percentage of cells bearing autophagosomes after treatment with pentamidine (10, 20 or 30 μM) or 17-AAG (500 nM) for 24 and 48 h. Lines within the floating bars represent medians and floating bar quartiles (Q: 25% and 75%) from one out of three independent experiments (Kruskal-Wallis test, Dunn’s multiple comparison test, * *p* < 0.05).

**Figure 2 microorganisms-09-01089-f002:**
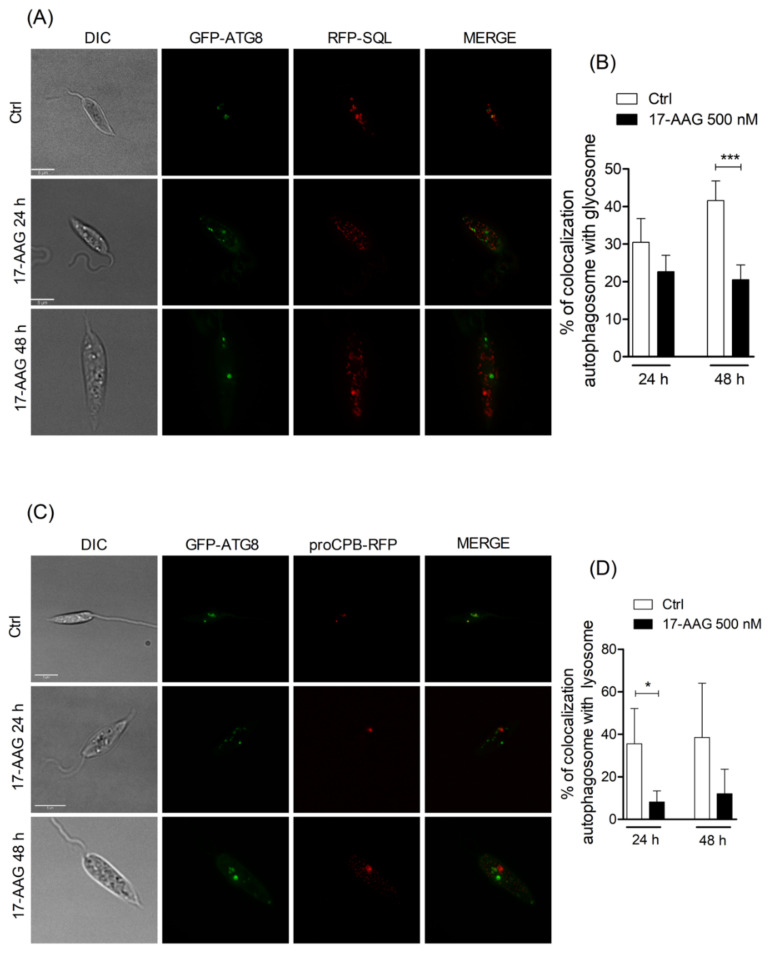
Analysis of fusion between autophagosomes and glycosomes or lysosomes. (**A**) Axenic promastigotes of *Leishmania* expressing GFP-ATG8 and RFP-SQL were treated or not with 17-AAG (500 nM) and imaged by fluorescence microscopy. (**B**) Quantification of autophagosome-glycosome colocalization after treatment of *Leishmania* with 17-AAG. (**C**) Axenic promastigotes of *Leishmania* expressing ATG8-GFP and proCPB-RFP were treated or not with 17-AAG (500 nM) and imaged by fluorescence microscopy. (**D**) Quantification of *Leishmania* autophagosome-lysosome colocalization after treatment with 17-AAG. Bars represent medians ± SD from one out of three independent experiments (Unpaired t test, *** *p* = 0.0006, * *p* = 0.0197).

**Figure 3 microorganisms-09-01089-f003:**
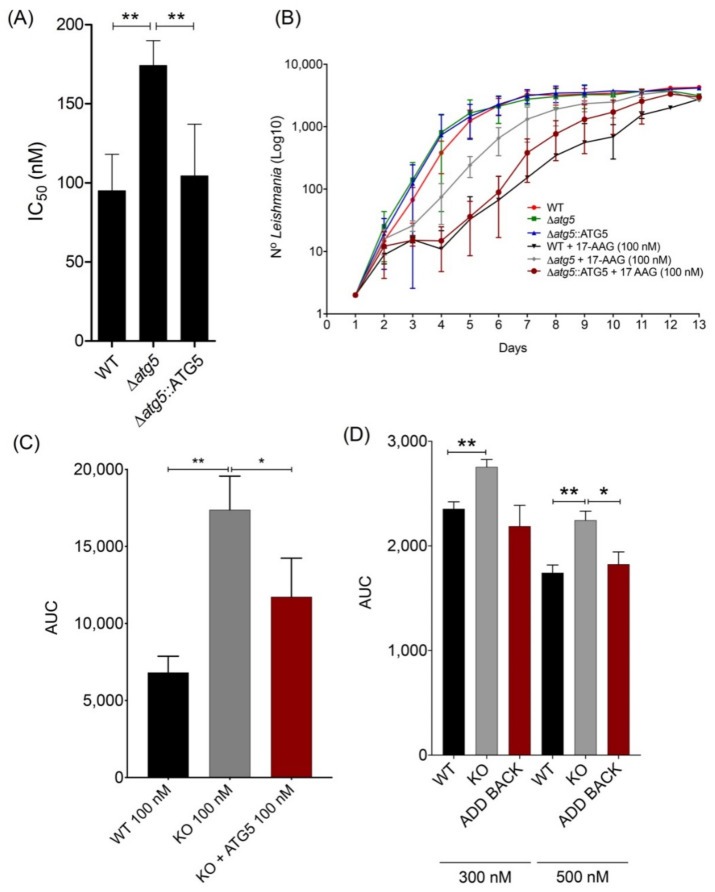
Effect of 17-AAG on survival and replication of WT, Δ*atg5* and Δ*atg5*::ATG5 *Leishmania*. (**A**) IC_50_ values for 17-AAG in different *Leishmania* lineages. Bars represent mean ± SD from four independent experiments (One-way ANOVA, Tukey’s multiple comparisons test, ** *p* < 0.01). (**B**) Growth curve reflecting 13 day-counts of WT, Δ*atg5* and Δ*atg5*::ATG5 parasites, treated or not with 17-AAG at 100 nM. Symbols are representative of means ± SD from three independent experiments. (**C**) Area under the curve (AUC) analysis of WT, Δ*atg5* and Δ*atg5*::ATG5 growth depicted in panel (**B**), following treatment with 17-AAG. Bars represent mean ± SD from three independent experiments (one-way ANOVA test, Tukey’s multiple comparison test * *p* = 0.0321 0.05, ** *p* = 0.0016). (**D**) AUC analysis of WT, Δ*atg5* and Δ*atg5*::ATG5 viability following treatment with 17-AAG at 300 nM and 500 nM for 24 h, 48 h and 72 h. Bars represent mean ± SD of a single experiment performed in quadruplicate (Welch’s ANOVA test, Dunnett’s T3 multiple comparison test * *p* < 0.05, ** *p* < 0.01).

**Figure 4 microorganisms-09-01089-f004:**
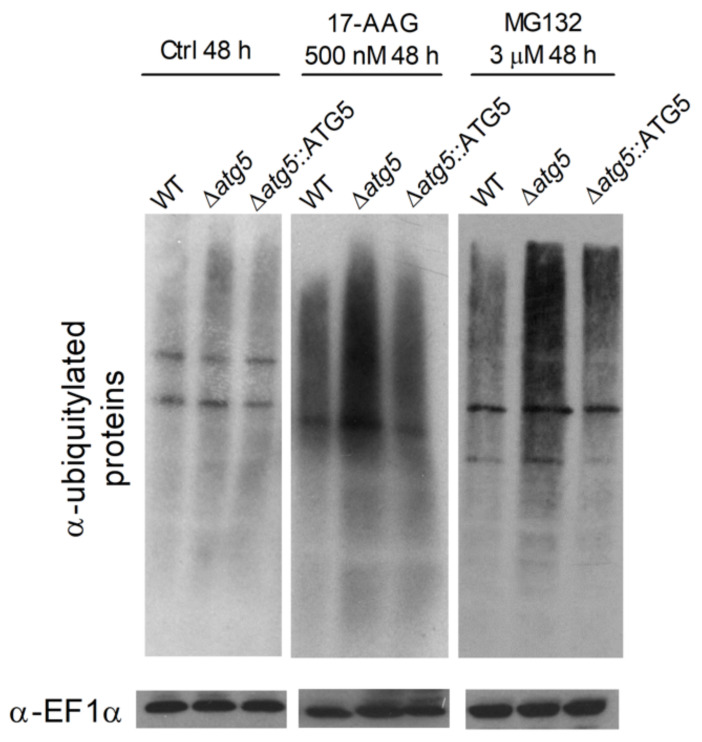
Ubiquitylated protein profiles of WT, Δ*atg5* and Δ*atg5*::ATG5 after 17-AAG or MG132 treatment. WT, Δ*atg5* and Δ*atg5*::ATG5 parasites were treated with 17-AAG (500 nM) or MG132 (3 µM) for 48 h. Protein extracts were electrophoresed on a 12% gel, blotted and probed with an FK2 anti-ubiquitin antibody. EF1α was used as loading control.

**Figure 5 microorganisms-09-01089-f005:**
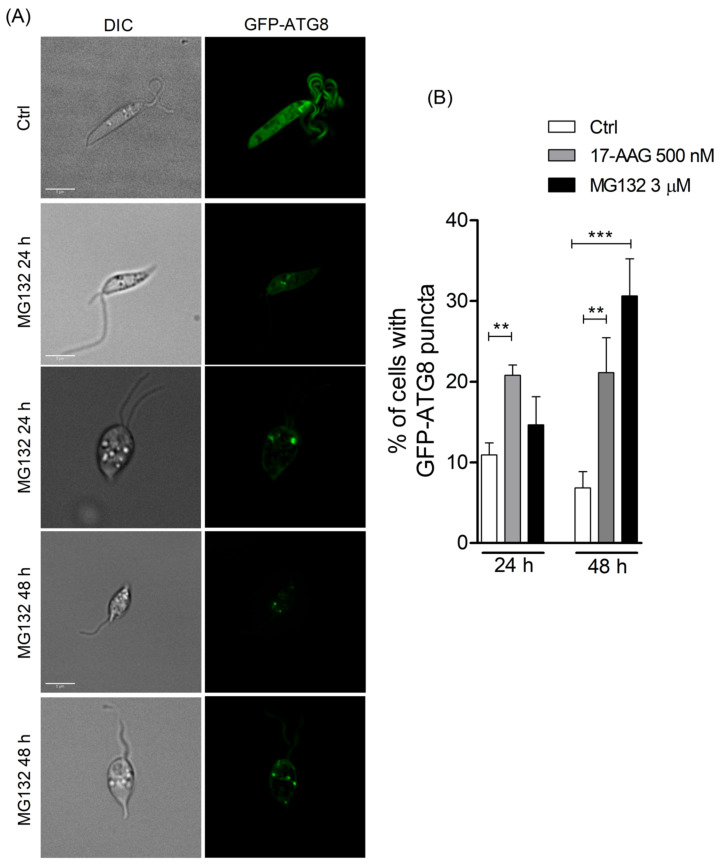
Evaluation of autophagosome formation in promastigotes of *Leishmania* following treatment with MG132. (**A**) Promastigotes of *Leishmania* expressing GFP-ATG8 were treated or not with MG132 (3 μM) and imaged by fluorescence microscopy. (**B**) The percentage of cells bearing autophagosomes was calculated after treatment with 17-AAG (500 nM) or MG132 (3 μM) for 24 and 48 h. Bars represent mean ± SD of three independent experiments (One-way ANOVA, Tukey’s multiple comparisons test, ** *p* < 0.01, *** *p* < 0.001).

**Figure 6 microorganisms-09-01089-f006:**
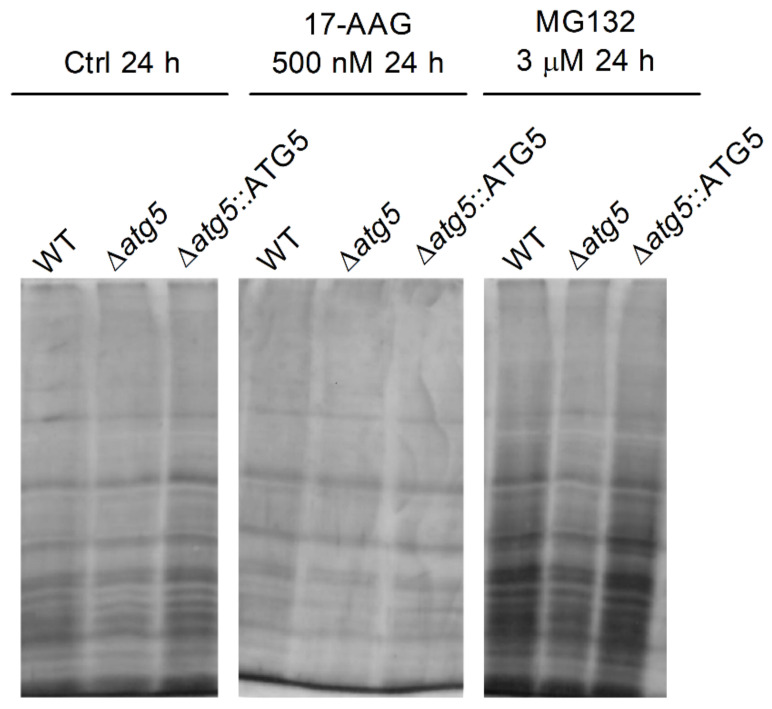
Effect of 17-AAG treatment on WT, Δ*atg5* and Δ*atg5*::ATG5 protein aggregate formation. Culture aliquots from WT, Δ*atg5* and Δ*atg5*::ATG5 parasites treated with 17-AAG (500 nM) or MG132 (3 µM) for 24 h were withdrawn and analyzed to determine quantities of insoluble protein aggregates by cell lysis and centrifugation. Protein aggregates were subjected to SDS-PAGE followed by silver staining. One experiment is representative of two independent experiments.

## Data Availability

Not applicable.

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
