# Peer review of "17-AAG-Induced Activation of the Autophagic Pathway in Leishmania Is Associated with Parasite Death"

_microorganisms, 2021, doi:10.3390/microorganisms9051089_

Round 1

Reviewer 1 Report

Review of 17-AAG-induced activation of the autophagic pathway in Leishmania is associated with parasite death.

Summary:

Leishmania is a very important neglected tropical disease for which new antiparasitic therapies are needed. Previous research has shown that 17-AAG inhibits Hsp90 and leads to parasite death, but its mechanism of action remains poorly understood and is the focus of the present study. The researchers hypothesized that 17-AAG induces abnormal activation of autophagy in Leishmania which ultimately results in parasite death. Using a set of L .major mutants, the researchers found evidence to support their hypothesis that 17-AAG causes Leishmania cell death. The main findings were that 17-AAG increases autophagy, reduces colocalization of autophagosomes with glycosomes or lysosomes, and that atg5 mutants had higher survival following exposure to 17-AAG compared to WT. Overall,  I thought the experimental design, analysis, and conclusions were all well designed, well written and analyzed and I believe this paper is suitable for publication, save for my comment regarding the figures.

Line 58-59:

Rephrase for clarity:

We have demonstrated that 17-AAG was capable of clearing Leishmania infection (in-vivo and in-vitro) by eliminating both promastigotes…..

Figures:

For ease of reading a comprehension, reduce the variation in the different types of figures used. Aim for the standard use of color (where it is appropriate) throughout all of your figures and use the same style of bar graph. Overall the figures are fine there is just no continuity from figure to figure (Some figures have bar graphs with each data point, with shades of grey, some graphs don’t have bars, some graphs don’t have data points) and the variability makes understanding each figure more difficult. What are you trying to express with color? Why do some graphs have color while others do not? Reduce the stylistic differences between figures to ease reader comprehension.

Author Response

According to reviewers ' comments, a marked-up copy detailing the changes made from the previous version has been prepared. We want to emphasize that criticisms have been carefully considered and that every amendment and clarification is also included in the authors’ responses here, separated by the referee.

The authors feel that the manuscript, in its present form, has greatly improved and we hope it is now suitable for publication in Microorganisms.

Reviewer comments: 

Reviewer #1 (Summary): 

Leishmania is a critical neglected tropical disease for which new antiparasitic therapies are needed. Previous research has shown that 17-AAG inhibits Hsp90 and leads to parasite death, but its mechanism of action remains poorly understood and focuses on the present study. The researchers hypothesized that 17-AAG induces abnormal activation of autophagy in Leishmania, ultimately resulting in parasite death. Using a set of L .major mutants, the researchers found evidence to support their hypothesis that 17-AAG causes Leishmania cell death. The main findings were that 17-AAG increases autophagy, reduces colocalization of autophagosomes with glycosomes or lysosomes, and that atg5 mutants had higher survival following exposure to 17-AAG compared to WT. Overall, I thought the experimental design, analysis, and conclusions were well-designed, well written, and analyzed. I believe this paper is suitable for publication, save for my comment regarding the figures.

Reviewer #1

Line 58-59: 

Rephrase for clarity: 

We have demonstrated that 17-AAG was capable of clearing Leishmania infection (in-vivo and in-vitro) by eliminating both promastigotes…..

Answer: The authors acknowledge the reviewer’s suggestion and have rephrased the sentence for clarity as follows.

(Lines 58-60) “In previous studies, we have demonstrated that 17-AAG was capable of controlling Leishmania infection (in vitro [15] and in vivo [16]) by eliminating promastigotes, which colonize the insect vector, as well as amastigotes, which are found within vertebrate host cells [15, 16].”

Figures: 

For ease of reading a comprehension, reduce the variation in the different types of figures used. Aim for the standard use of color (where it is appropriate) throughout all of your figures and use the bar graph style. Overall the figures are fine there is just no continuity from figure to figure (Some figures have bar graphs with each data point, with shades of grey, some graphs don’t have bars, some graphs don’t have data points) and the variability makes understanding each figure more difficult. What are you trying to express with color? Why do some graphs have color while others do not? Reduce the stylistic differences between figures to ease reader comprehension.

Answer: The authors acknowledged the reviewer`s suggestion and for improving reading comprehension, we revised the format of the figures throughout the manuscript and statistical analyses. The graphs in figure 1 were kept unaltered since medians and floating bars and quartiles accurately represent the nonparametric results shown in these graphs (Fig. 1B and Fig. 1D).

Fig. 2C and 2D have been modified from dots to bars (mean ± SD). A review of our results revealed that data presented herein present parametric distribution; thus, the appropriate test to be used is the unpaired t-test. Subtitles have been properly modified in accordance.

Fig 3A was changed from bars plus dots to solid black bars representing mean ± SD. These results have a parametric distribution; accordingly, statistical analysis has been redone using one-way ANOVA, followed by Tukey's multiple comparisons test. Figs 3B, 3C and 3D have color scheme modified. The authors would like to point out that the color selection from Fig. 3C and 3D is meant to meet the same color scheme from Fig. 3B. The Area Under the Curve from Fig. 3C is based on the experimental groups (WT + 17-AAG 100 nM; Δatg5 + 17-AAG 100 nM; Δatg5::ATG5 + 17-AAG 100 nM) shown at Fig. 3B.

Figure 5B was also changed from bars plus dots to bars representing mean ± SD. Data showed a parametric distribution; subsequently, a nonparametric statistical analysis has been done, using one-way ANOVA, followed by Tukey's multiple comparisons test.

Reviewer 2 Report

The manuscript entitled “17-AAG-induced activation of the autophagic pathway in Leishmania is associated with parasite death” presents interesting results about the mechanism of action of 17-AAG that merit publication in the journal Microorganisms. However, some comments need to be addressed.

  1. Use italics for in vitro and in vivo (abstract, lines 48, 59, ….)
  2. Revise the entire manuscript to correct errors (500 nM instead of 500nM in line 329, Figure 1: 500nM, 48h, 30µM…).
  3. In Figure 1A (page 6), there are 2 photos of 17-AAG 24 h and 2 photos of 17-AAG 48 h. Are these photos the same or one is taken after 300 nM and the other after 500 nM treatment? If yes, add the concentration in the photos to clarify. The same in Figure 2A and 2C with 24 h and 48 h and Figure 5A. Include time or concentration where corresponds.
  4. In Figure 4 and Figure 6, what is the concentration of 17-AAG and MG132 used? Please, include this information in the figures and in the legends.
  5. Have authors check same autophagy markers (LC3B, p62) after 17-AAG treatment? Are they upregulated?

Author Response

According to reviewers ' comments, a marked-up copy detailing the changes made from the previous version has been prepared. We want to emphasize that criticisms have been carefully considered and that every amendment and clarification is also included in the authors’ responses here, separated by the referee.

The authors feel that the manuscript, in its present form, has greatly improved and we hope it is now suitable for publication in Microorganisms.

Reviewer #2

The manuscript entitled “17-AAG-induced activation of the autophagic pathway in Leishmania is associated with parasite death” presents interesting results about the mechanism of action of 17-AAG that merit publication in the journal Microorganisms. However, some comments need to be addressed.

  1. Use italics for in vitro and in vivo (abstract, lines 48, 59, ….)

The authors acknowledge the reviewer`s suggestion and agree that it is more precise to use italics to write in vitro and in vivo, however following item 3.6 from MDPI Style Guide (https://www.mdpi.com/authors/layout#_bookmark15) reads: “Foreign words do not need to be highlighted or italicized, including Latin terms such as ‘in situ.” in vivo and in vitro should remain in its non-italic form.

  1. Revise the entire manuscript to correct errors (500 nM instead of 500nM in line 329, Figure 1: 500nM, 48h, 30µM…).

The authors thanks reviewer perception and corrected these errors throughout the entire manuscript, including figures.

  1. Figure 1A (page 6) shows 2 photos of 17-AAG 24 h and 2 photos of 17-AAG 48 h. Are these photos the same or one is taken after 300 nM and the other after 500 nM treatment? If yes, add the concentration in the photos to clarify. The same in Figure 2A and 2C with 24 h and 48 h and Figure 5A. Include time or concentration where corresponds.

The authors agree with the reviewer that more clarity is needed in all figures of the manuscript. Regarding figure 1A all images correspond to cell treatment with 500 nM of 17-AAG. For clarity, 17-AAG concentration was included in the legend of the figure. In figures 2A and 2C, only one concentration was used to treat parasites (500 nM), so this information was included in the figure legend.

  1. In Figure 4 and Figure 6, what is the concentration of 17-AAG and MG132 used? Please, include this information in the figures and the legends.

The authors acknowledge the reviewer`s suggestion and added the information requested.

  1. Have authors checked the same autophagy markers (LC3B, p62) after 17-AAG treatment? Are they upregulated?

We have not checked autophagy markers other than ATG8, which is the known autophagic marker expressed in Leishmania parasites, closely related to the human LC3 marker. A cell-line expressing GFP-ATG8 is largely employed for monitoring autophagy in Leishmania [1, 2, 3]. Using the same L. major cell-line, it was previously shown an increase in GFP-ATG8 lipidation directly correlated with enhancement in the number of GFP-ATG8 puncta structures [3]. To our knowledge, p62 is a known autophagy adapter expressed only in mammalian cells generally used as a selective autophagic marker. 

1 -  Guidelines for the use and interpretation of assays for monitoring autophagy (3rd edition). Klionsky DJ et al. Autophagy. 2016;

2 -  Glycosome turnover in Leishmania major is mediated by autophagy.

Cull B, Prado Godinho JL, Fernandes Rodrigues JC, Frank B, Schurigt U, Williams RA, Coombs GH, Mottram JC. Autophagy. 2014;

3 - Characterization of unusual families of ATG8-like proteins and ATG12 in the protozoan parasite Leishmania major. Williams RA, Woods KL, Juliano L, Mottram JC, Coombs GH. Autophagy. 2009